# Self-strengthening biphasic nanoparticle assemblies with intrinsic catch bonds

Kerim C. Dansuk[1] & Sinan Keten [1,2✉]

Protein–ligand complexes with catch bonds exhibit prolonged lifetimes when subject to tensile force, which is a desirable yet elusive attribute for man-made nanoparticle interfaces and assemblies. Most designs proposed so far rely on macromolecular linkers with complicated folds rather than particles exhibiting simple dynamic shapes. Here, we establish a scissor-type X-shaped particle design for achieving intrinsic catch bonding ability with tunable force-enhanced lifetimes under thermal excitations. Molecular dynamics simulations are carried out to illustrate equilibrium self-assembly and force-enhanced bond lifetime of dimers and fibers facilitated by secondary interactions that form under tensile force. The non-monotonic force dependence of the fiber breaking kinetics is well-estimated by an analytical model. Our design concepts for shape-changing particles illuminates a path towards novel nanoparticle or colloidal assemblies that have the passive ability to tune the strength of their interfaces with applied force, setting the stage for self-assembling materials with novel mechanical functions and rheological properties.

[1] Department of Mechanical Engineering, Northwestern University, 2145 Sheridan Road, Evanston, IL 60208, USA. [2] Department of Civil & Environmental Engineering, Northwestern University, 2145 Sheridan Road, Evanston, IL 60208, USA. ✉email: s-keten@northwestern.edu

Nanoparticles can assemble into larger, microscale structures such as sheets, tubes, wires, and shells, under conditions of anisotropic interactions[1]. The shape of common nanoparticles is typically static, dictating the hierarchy of assembly[2,3], while also influencing the binding forces that hold particles together[4], which consequently governs the mechanical properties of the emergent structures[5]. Nature has a plethora of building blocks with unique shape-performance relations[6]; however, unlike most of their synthetic counterparts, biological building blocks, e.g., proteins, can undergo dynamic shape changes under variations of pH, temperature changes, solvent composition alterations, and mechanical deformation. The multiphasic shape features of proteins facilitated by conformational dynamics control binding kinetics. This is highly desired in colloids and nanoparticles, but existing methods to do this typically rely on active stimuli such as temperature or light-responsive polymers.

Biomimetic nanoparticles that undergo functional shape changes are critically needed to create self-assembling materials with force-tunable interfacial kinetics. This work aims to design self-assembling anisotropic nanoparticles that can undergo reversible shape changes inspired by catch bonds in protein–ligand complexes, which are incorporated in biosensing, adhesion, and many other functions of biological systems. These complexes utilize allosteric regulation of the binding site, resulting in a counterintuitive increase in the lifetime of the complex when force is applied to the ligand[7]. This unusual form of binding was demonstrated in several proteins: P-selectin/L-selectin[8], pili adhesin FimH[9], integrin[10], actin[11], von Willebrand factor[12], and kinetochore[13], all of which reversibly bind to their respective ligand and resist removal under high stress by strengthening the protein–ligand interaction.

Following the proof of their existence at a single-molecule level for a significant number of proteins, catch bonds are now receiving attention from the material science community with the intention of utilizing their strengthening mechanism for material interfaces in architected nanoparticle networks[14]. Given that catch bonds are reversible interactions that become stronger with applied force, they appear to be excellent candidates both for providing resistance to deformation and for enabling reconfiguration in material systems. While catch bonds have been shown theoretically and computationally[15–17] to enhance mechanical properties of network nanocomposites, and certain nonbiological molecules seem to exhibit catch bond behavior[18,19], integration of catch bonds in materials has not yet been realized as structures with highly tunable catch bond-like characteristics have not yet been fabricated. So far, the catch bond research has led to the development of several theoretical models[20,21]. Phenomenological models, such as the two-state model[22,23] and the two-path model[24,25], can accurately represent characteristics of a wide range of experimental data sets, but provide limited information on how one should create a structure and mechanism for catch bonds, since the parameters extracted from these models do not have clear structural interpretations.

Our approach to this problem is to design simple mechanical systems that can capture (i) structural changes in catch bond proteins due to mechanical forces, and (ii) alterations in unbinding kinetics due to structural changes. Prior experiments and simulations[20,26–28] have shown that some catch bond proteins, including FimH, selectin, and cadherin, are biphasic, consisting of a conformation with low ligand-binding affinity and a conformation with high ligand-binding affinity. Tensile force applied to the ligand increases the probability of a transition from the low- to high-affinity conformation, which leads to a prolonged lifetime at intermediate levels of tensile force. From these observations, we identified three key features of catch bond proteins: (i) existence of two discrete conformational states, (ii) force-induced transition between the states, and (iii) increase in ligand affinity by formation of new interactions between the protein and the ligand due to structural changes that occur during transitions between states.

In our previous work, we explored these key features and developed a simple tweezer-like design that exhibits catch bond behavior when simulated under thermal excitations[29]. Here, we will utilize the insights from the tweezer structure to design biphasic nanoparticles that can self-assemble into 1D fibers, which constitutes a demonstration of the self-assembly of units with intrinsic catch bonds, and opens new questions about the scaling of strength in such systems. Building on the success of the tweezer model that forms additional contacts upon application of force, here we pursue a scissor-type X-shaped nanoparticle design that can form intrinsic catch bonds with itself. We utilize molecular dynamics (MD) simulations to show that the nanoparticles can self-assemble into fibers with force-enhanced lifetimes. Simulations and theoretical considerations are utilized to explain the catch bond kinetics in dimers, and the scaling of strength in larger oligomeric assemblies. These results are followed by a discussion on the relevance of these findings to make synthetic catch bonds and a direct link to DNA origami systems that can currently be synthesized.

We believe that the proposed structure and mechanism for catch bonds is a crucial intermediate step for the inverse problem of synthetic design. Unlike the phenomenological models in the literature, our system does not rely on experimental data to train a model and makes no assumptions on energy landscapes or bond-breaking kinetics. Lifetime curves are directly obtained from MD simulations. In addition, our model displays (i) explicit structures for two conformational states, (ii) a switch mechanism to smoothly transition between states with real kinetics, and (iii) complementary interaction sites that allow self-assembly of arbitrarily long fibers, which are all structural features that are needed for synthetic design, but were not captured by the earlier frameworks. Furthermore, the molecular model is capable of self-assembly, which paves the way for scalability of catch bond-based systems.

## Results

**Structure design.** To achieve intrinsic catch bonds between particles, we propose a design that incorporates an X-shaped nanoparticle with a hidden binding site that becomes exposed upon application of tensile force, and a switch that controls the likelihood of this conformational transition. The X-shaped nanoparticle is a symmetric structure consisting of rigid members that are connected with hinge joints as shown in Fig. 1a. The nanoparticle has four binding sites at the ends of its arms (red beads) and two binding sites along the central vertical axis (blue beads). In this work, we will refer to red beads as primary binding sites (PBS) and the blue beads as secondary binding sites (SBS). Gray beads have steric repulsion with all the beads to create inaccessible volume determining the shape of the nanoparticles, which influences the morphologies formed via self-assembly by providing shear resistance to the fiber and enabling correct attachment during the self-assembly process. Arms of the structure can rotate about the center; thus, the two halves of the structure have open and closed conformational states. It should be noted that there is no angle constraint between the halves; thus, they can open and close independently from each other.

To better understand how X-shaped nanoparticles interact with each other in different conformational states, we focus on a preassembled dimer. As shown in Fig. 1b, when nanoparticles are in the open conformation, complementary PBS is connected by

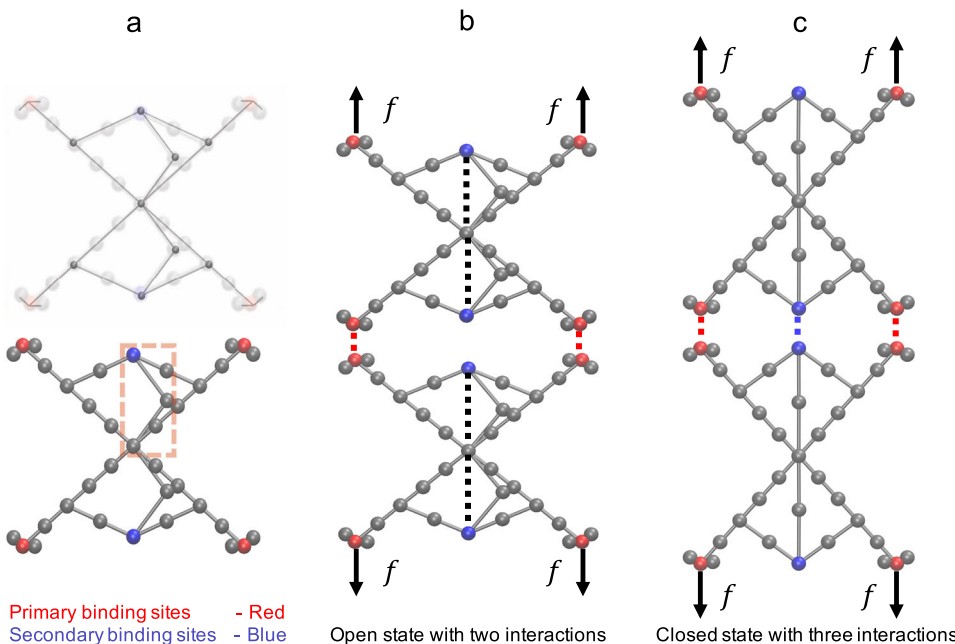

**Fig. 1 Schematics of X-shaped nanoparticles. a** Each nanoparticle is formed by rigid members connected with hinge joints (gray nodes). Each nanoparticle has primary binding sites (PBS, red), secondary binding sites (SBS, blue), and switch mechanisms. The hinge-like switch is highlighted with a dashed box. Gray beads have steric repulsion to create inaccessible volume determining the shape of the nanoparticles. **b** When two nanoparticles are in the open state, their switch members are angled and they can form a dimer via two intermolecular interactions (red dashed lines). **c** When two nanoparticles transition to the closed state, the switch members become straight and the dimer forms an additional intermolecular interaction (blue dashed line). The switch interactions and applied external forces are indicated by black dashed lines and black arrows, respectively.

nonbonded interactions (red dashed lines). In this equilibrium-bound state, SBS interaction is negligible, since the intermolecular distance between SBS is large. The force-dependent transition between the two conformations is controlled by switches in both the upper and lower halves of the nanoparticle. The switch consists of a two-member hinge with a pairwise interaction between its two free ends (blue SBS bead and black bead at the symmetry axes). In the open conformation, the switch members are angled due to the pairwise interactions as shown with black dashed lines in Fig. 1b. However, these pairwise interactions can break due to external forces. In that case, since there is no other force between the members, the system elongates in the direction of the force and the switch members straighten and become linear (Fig. 1c). Straightening of the switch moves the SBS upward and drives the arms inward, enabling interaction between the complementary SBS (blue dashed line). The new interaction increases the total binding energy between the nanoparticles, giving the dimer a higher affinity in the closed conformation. Thus, in the context of unbinding, dimers in the open conformation are in their low-affinity (LA) state and those in the closed conformation are in their high-affinity (HA) state. In conclusion, these X-shaped nanoparticles have all three key catch bond features established earlier: they have two (open and closed) conformations, force regulates the breaking and reforming of switch interactions controlling the transition between the conformations, and SBS interaction formed upon transition to the closed conformation increases the affinity of the particles. Detailed description of the nanoparticle system is presented in the Supplementary Information, including the masses and the dimensions shown in Supplementary Fig. 1.

Depending on the conformation of the nanoparticles forming the dimer, four events can be observed. In the LA state, the dimer may undergo LA unbinding, or it may transition to the HA state. In the closed conformation, the dimer may undergo HA unbinding, or connected halves may transition to the open

conformation. These four events occur in a stochastic fashion due to the presence of thermal motion. Therefore, transition/unbinding events have no specific lifetime, but rather follow a distribution of lifetimes. To characterize the probability of these events as a function of force and obtain the lifetime distributions, multiple trials need to be performed at each force magnitude. Average lifetimes $\langle\tau\rangle$ can then be computed from these lifetime distributions.

**Binding energy landscapes for catch bond behavior.** In our study, Morse potentials are used to represent the PBS interactions ($E_{PBS}$), SBS interactions ($E_{SBS}$), and the interactions that make up the switch ($E_S$) with the general form

$$E(x) = D_0(\exp[-2\alpha(x - x_o)] - 2\exp[-\alpha(x - x_o)]). \quad (1)$$

Here, $D_0$ is the depth of the energy well, $x_o$ is the equilibrium bond distance, and $\alpha$ is the parameter that controls the width of the well (the smaller $\alpha$ is, the broader the well). An advantage of the Morse potential over the more common Lennard–Jones potential is the ability to control binding energy and landscape curvature independently. To reproduce the characteristic catch bond lifetime curve, the mean lifetime of the dimer $\langle\tau\rangle_2$ must increase when a tensile force is applied. In our previous work with a stationary mechanical model[29], we have identified two conditions that result in the force-enhanced lifetimes seen in catch bonds. First, at small force, most trials should result in LA unbinding of the dimers, and at large forces, most trials should transition to the HA state before LA unbinding can occur. Second, the mean HA unbinding lifetime $\langle\tau\rangle_{HA}$ must be longer than the mean LA unbinding lifetime $\langle\tau\rangle_{LA}$. Satisfying these two conditions will result in a higher percentage of trials unbinding at larger lifetimes, which guarantees the increase in $\langle\tau\rangle_2$ as the pulling force increases.

To satisfy the first condition, we determined that the switch should have a deeper and a broader energy landscape than the binding site landscape[29]. At zero or small forces, the switch lifetime should be longer than the LA unbinding lifetime to have the majority of the pulling trials result in dimers separating in the LA state before transitioning to the HA state. According to Kramer's theory[30], $\langle \tau \rangle$ is directly proportional to exp$[D_0]$; thus, by selecting the depth of the energy well of the switch ($D_{0,S}$) to be greater than depth of the energy well of the PBSs ($D_{0,PBS}$), we obtain a higher switch lifetime at small forces. Note that in catch bonds, conformational changes occur over relatively long time-scales in the absence of force, which is also why they are difficult to observe with MD simulations[9,31]. This suggests that there is a large barrier to these conformational changes, which is in qualitative agreement with our choice for $D_{0,S}$. On the other hand, the LA unbinding lifetime should be longer than the switch lifetime to have majority of the pulling trials result in HA-state transition at large forces. This transition between the relative lifetimes will be controlled via tuning the width of the energy landscapes of the system. Note that the tilting of the energy landscape by force is sensitive to the distance to the transition state, or the width of the landscape when well depth is kept constant. Therefore, $E_S$ must have lower $\alpha$ (broader well) compared to $E_{PBS}$, which will result in the energy barrier of the switch landscape to decrease rapidly when greater tensile force is applied. Hence, the switch lifetime declines more sharply when subjected to force compared to the LA unbinding lifetime. Based on this condition, we have determined the parameters for $E_S$ and $E_{PBS}$, which are listed in Table 1. For an improved understanding of this response, we performed molecular dynamics simulations of tensile pulling of the dimer. The simulations were terminated (i) when the particles dissociate (LA unbinding event) or (ii) when the switch interactions dissociate (LA- to HA-state transition event). Figure 2 shows the fraction of these two events at various forces. At low forces, the majority of the trials result in dimers separating in the LA state before the HA-state transition, e.g., 90% of the trials at 140 pN. This fraction decreases monotonically as the constant pulling force is increased, e.g., to 3% at 280 pN. The decrease from 90 to 3% agrees with the first catch bond condition listed above.

The second condition we have specified is based on the notion that the formation of the SBS interaction increases the unbinding energy barrier. Since the unbinding barrier is higher, $\langle \tau \rangle_{HA}$ will be greater than $\langle \tau \rangle_{LA}$. In the following simulations, we perform tensile pulling on nanoparticle dimer to analyze the effect of secondary interaction ($E_{SBS}$) on dimer lifetime. We select $\alpha$ and $x_o$ of $E_{SBS}$ to be the same as the primary interaction parameters, but varied $D_{o,SBS}$ between 0 and 1 kcal/mol. The simulations were terminated when the particles dissociate (LA and HA unbinding event) and dissociation lifetimes of the dimers are recorded. Animation of sample simulation runs is provided as Supplementary Movie 1 and Supplementary Movie 2. As seen in Fig. 3, in the absence of secondary interactions ($D_{o,SBS} = 0$), the lifetime of the dimer exponentially decreases with force (dashed line), which is expected for typical chemical interactions/bonds[32]. For all the nonzero $D_{o,SBS}$ cases, we use peak lifetime $\langle \tau \rangle_{peak}$ and the corresponding critical force $f_c$ to characterize the lifetime

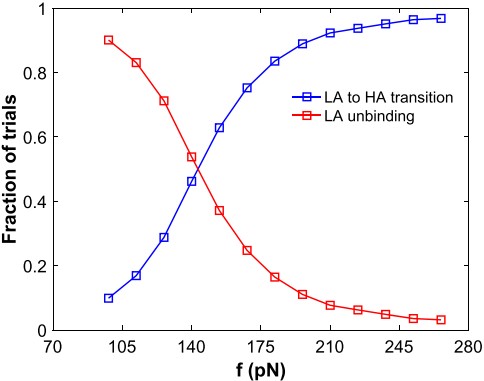

**Fig. 2 Force dependence of the fraction of trials for the dimer to dissociate in the LA state and dimer to transition from the LA to the HA state.** LA unbinding is shown in red and LA-to-HA transition is shown in blue.

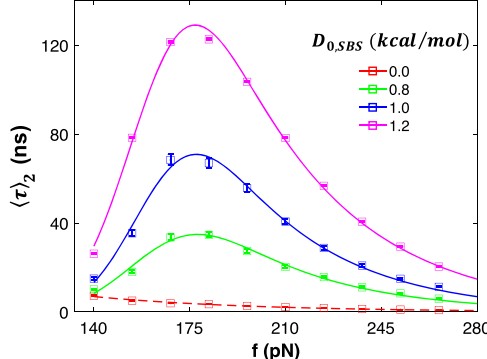

**Fig. 3 Effects of secondary interaction energy well depth ($D_{o,SBS}$) on mean lifetime $\langle \tau \rangle$ vs. force $f$ curve of a nanoparticle dimer.** Symbols are from MD simulations and the curves are generated using an exponentially modified Gaussian distribution fitting function. The error bars represent the 95% confidence intervals, derived by the bootstrapping method.

simulation results. Figure 3 shows that $\langle \tau \rangle_2$ of all the solid curves has peaks around $f_c = 175$ pN. After the peak, $\langle \tau \rangle_2$ monotonically decreases at larger forces. As expected, when the $E_{SBS}$ is increased, the overall lifetime for the dimer increases, i.e., the higher the secondary interaction energy, the higher the increase in unbinding the energy barrier.

Hd LA unbinding pathways described above can be represented as a multidimensional energy landscape that depends on interparticle distance $x$ and intraparticle angle $\theta$. Figure 4a displays selected snapshots from the simulations that show the unbinding (green, orange) and switch (black) pathways. Since the upper and lower halves of the nanoparticle can rotate independently, interaction of the dimer pair will be governed by the halves forming the interface. Thus, two switches, each having energy of $E_s$, control the change in $\theta$. The transition energy landscape (black curve in Fig. 4b) has two local minima, demonstrating the biphasic nature of the dimer. When $\theta$ decreases to 42°, switch members are straight and any force to deform the nanoparticle will be carried by these members, which are fairly rigid and thus prohibit further variations in the geometry. Deforming the system beyond this point requires a large energy input; thus, there is a big surge in t transition energy below 42°. Similarly, above 50°, since the distance between switch elements is less than the Morse interaction equilibrium distance of $E_S$, the short-range repulsion term of the potential dominates, and we see an increase in energy. Moreover, the force-induced

**Table 1 Interaction parameters.**

| Interaction | $D_O$, kcal/mol | $\alpha$, Å$^{-1}$ | $x_o$, Å |
|---|---|---|---|
| Primary, $E_{PBS}$ | 2.8 | 10 | 1.5 |
| Switch, $E_S$ | 3.4 | 3 | 3.5 |

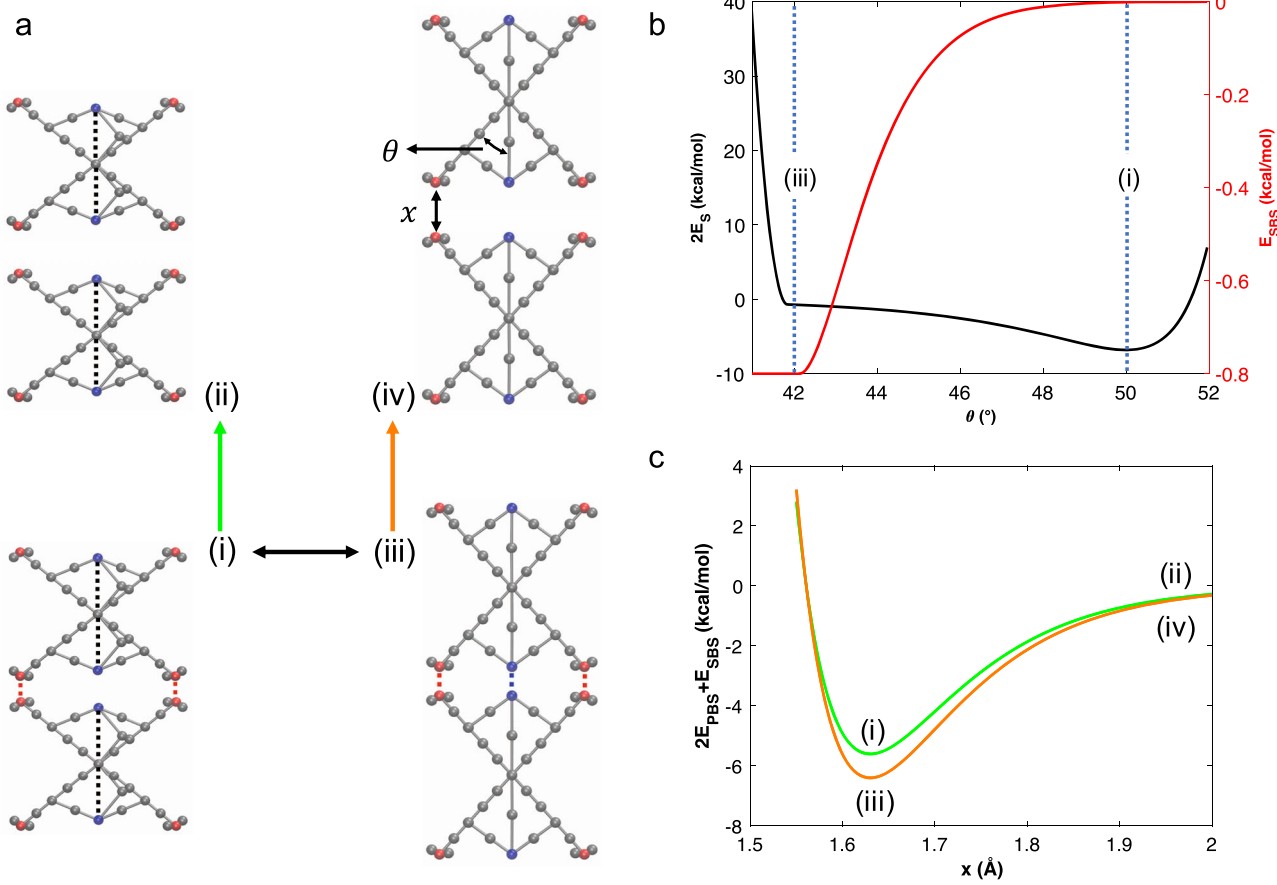

**Fig. 4 X-shaped nanoparticles form a biphasic system with two unbinding pathways. a** Snapshots (i) and (iii) are the LA state and HA state, respectively. Snapshots (ii) and (iv) are the unbound dimers. The black line indicates the transition pathway of the dimer between its LA and HA states. The green line is the LA unbinding pathway and the orange line is the HA unbinding pathway. The reaction coordinates, namely interparticle distance $x$ and intraparticle angle $\theta$, are marked in snapshot (iv). **b** Energy landscapes for the LA-to-HA-state transition as a function of intraparticle angle $\theta$. The transition energy landscape (black curve) is between the complementary halves of two nanoparticles and is the sum of two switch interactions. $E_{SBS}$ (red curve) shows that in the LA state, SBS has no contribution to the interaction between the particles; however, in the HA state complementary SBSs interact. **c** Energy landscapes for the LA to HA unbinding as a function of interparticle distance $x$. The curves follow the same color code as the pathways. HA unbinding has a larger barrier due to $E_{SBS}$ contributions.

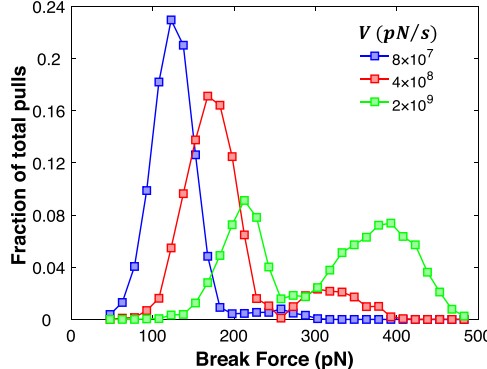

**Fig. 5 Break force histograms for X-shaped nanoparticle dimer under constant loading rate.** The data points are expressed as fractions of total pulls (10,000 trials) for each of the three loading rates $V$ (blue, red, and green).

decrease in $\theta$ brings SBS closer and a new interaction is formed (red curve). The binding energy between the particles is defined as $2E_{PBS} + E_{SBS}$, incorporating the three possible interactions between the particles. The energy landscapes for the unbinding pathways are shown in Fig. 4c. Since $E_{SBS}$ increases in the HA

state, HA unbinding has a higher energy barrier than LA unbinding.

In the end, the two conditions we specified result in catch bond behavior at the nanoparticle interface. At small forces, most trials unbind in the LA state, exhibiting short lifetimes. At larger forces, the unbinding is delayed because more trials switch to the HA state with the associated additional interactions and the backward transition becomes less likely, which prolongs $\tau_2$ of the individual trials. It should be noted that the temperature and the force ranges used in the simulations are arbitrary and do not aim to reproduce any particular experimental setup. Our goal here is to demonstrate the catch bond behavior in generic nanoparticle systems. As long as the two conditions we have specified are satisfied, it is possible to tune the system to exhibit a lifetime peak at arbitrary forces[29].

**Force history dependence of the system**. Since the dimer can switch between and unbind at different conformations depending on the instantaneous value of the force, the catch bond lifetime depends on the loading history. Indeed, the effects of loading history on catch bond lifetimes are investigated in FimH and P-selectin proteins via force-ramp experiments, where a probe is used to pull apart the protein–ligand pair under constant loading rate[33,34]. These experiments show that protein–ligand

interactions break at low or high forces, but are the strongest at an intermediate range of force. It is also observed that as the loading rate is increased, the breaking events occur more frequently at higher forces; thus, at faster loading rates, low-force rupture events are less likely to occur. To investigate these effects, we performed simulations where the tensile force on the dimer is increased at a constant loading rate, and we record the magnitude of the force when the bond breaks, $F_b$. The histogram of $F_b$ for different loading rates is shown in Fig. 5. The curves exhibit a bimodal distribution, with one peak at low forces and another at high forces. This shows that, for our system, the majority of the pulling trials break at low and high forces, but not at in-between intermediate range, which is in agreement with force-ramp experiments. Similar to the experiments, we observe that as we increase the loading rate, we observe an increase in the high force peak in the histogram and decrease in the low-force peak. Last, overall curves shift to the left with the increasing loading rate. This is expected since bond strengths are generally increased by the loading rate[35].

It should be noted that the simulation loading rates are five orders of magnitude higher than the experimental loading rates of single-molecule force spectroscopy experiments on catch bond systems. In our system, due to computational time restrictions, we have chosen the energy barriers of the particle interactions to be lower than the energy barriers of protein–ligand complexes to efficiently sample bond-breaking events. Moreover, to reduce the noise, these simulations are performed at a lower temperature than the single-molecule force spectroscopy experiments. As a result, our event lifetimes are significantly shorter than catch bond lifetimes seen in experiments. Consequently, higher loading rates have to be used to reach the critical force governing low-to-high-affinity transition.

**Self-assembly of nanoparticles.** In order to investigate the self-assembly of the nanoparticles, we have generated unbiased random initial configurations for 100 open-conformation particles and ran simulations to sample fiber formation trajectories. The replica-exchange method is used to perform the self-assembly simulations, which enables us to increase the effective simulation time of MD simulations by improving sampling of the relevant areas of the assembly path. In the simulations, nanoparticles interact from PBS and successfully form fibers with lengths ranging from 2 to 10 particles. None of the nanoparticles transition to closed conformation because the timescale of the open-to-closed-conformation transition in the absence of force is remarkably longer than unbinding and self-assembly lifetimes. A sample simulation trajectory is shown in Supplementary Movie 3. Note that during the trajectory, the simulation temperature varies over time as swaps take place between replicas.

**Lifetime scaling of self-assembled fibers.** With the aim of determining whether the catch bond behavior that we observed in dimers is retained in longer self-assembled fibers, we run tensile pulling simulations of fibers with various numbers of particles. For the simulations, $D_0$ of the $E_{SBS}$ is chosen to be 0.8 kcal/mol. Figure 6 shows that catch bond behavior is present regardless of fiber length. There are lifetime peaks at $f_c = 175\ pN$ for all lengths, even though $\langle \tau \rangle_{peak}$ of the fiber decreases as the number of subunits increases. The following observation can be explained with probability theory considering a serial arrangement of interactions. The probability that a dimer exposed to a constant force $f$s still intact after time t is given by

$$P_2(F) = e^{-r(f)t}, \tag{2}$$

where $r\ (f)$ is the rate of dissociation of the dimer. The expected

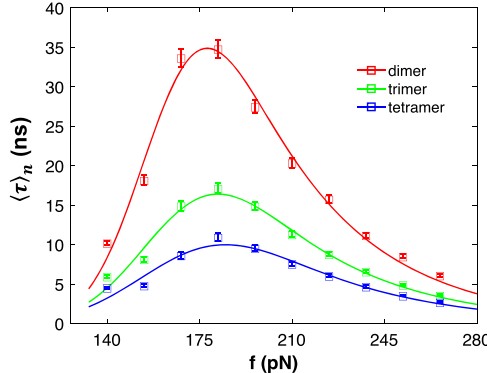

**Fig. 6 Mean lifetime $\langle \tau \rangle$ vs. force $f$ curves for self-assembled fibers at varying degrees of polymerization.** Symbols are from MD simulations and the curves are generated with an exponentially modified Gaussian distribution fitting function. The error bars represent 95% confidence intervals, derived by the bootstrapping method.

lifetime can be calculated using

$$\tau = -\int_0^\infty \frac{dP}{dt} t\, dt. \tag{3}$$

At a constant pulling force, $r$ will be constant; thus, the lifetime of the dimer $\langle t \rangle_2$ is $1/r$ using Eq. 3. Note that $r$, $P$, and $\langle t \rangle$ are all functions of pulling force. Similarly, we can calculate the probability of the three-particle fiber (trimer) being intact, $P_3$. Here, we make two assumptions. First, the dissociation events (unbinding of particles 1, 2, or 3) are independent from each other. Therefore, $P_3$ will be a pr.dt of the probability of interaction between the first and the second subunit being intact and probability of interaction between the second and the third subunit being intact. Second, the force along the fiber, i.e., tilting on all the energy landscapes, is the same. Therefore, the probability of unbinding is the same for all subunits, giving us the relation

$$P_3 = P_2 \cdot P_2 = e^{-2rt}. \tag{4}$$

Using Eq. 3, the lifetime of the trimer is found to be

$$\langle \tau \rangle_3 = \frac{1}{2r} = \frac{\langle t \rangle_2}{2}. \tag{5}$$

The following relation can be expanded for n-mer structure; therefore, the expected lifetime of n-mer fiber will be

$$\langle \tau \rangle_n = \frac{\langle t \rangle_2}{n-1}. \tag{6}$$

With $\langle \tau \rangle_2$ known, we predict the lifetimes of trimer and tetramer fibers using Eq. 6, which match with the curves in Fig. 6. Thus, we have shown that we can predict the fiber lifetime at any arbitrary degree of oligomerization.

## Discussion
Inspired by the mechanical behavior of biological catch bonds, we developed a model to probe how introduction of such catch bonds into nanoparticles could lead to fibers with intrinsic self-strengthening properties when subject to extreme stresses. Our analysis proves that nanoparticles with intrinsic catch bonds will exhibit force-enhanced lifetimes when self-assembled into fibers, and in dimer state exhibit stronger binding under larger mechanical stresses. Shape-changing, or shape-shifting nanoparticles are mostly used in medical applications, such as drug delivery, and their potential in mechanical applications remains to be realized. This work is a demonstration of not only a shape-

changing nanoparticle, but also a synthetic catch bond system as a new class of building block for the fabrication of ordered structures. While the structure we study is specific, the concept and our analysis methodology are broadly applicable to assemblies where buckling, snap-through, or breaking of weak internal bonds (e.g., in foldamers) can facilitate shape changes that alter interparticle interactions. We hope that this work serves as a starting point for synthetic chemists aiming to design such architected nanoparticles to generate emergent functional materials through self-assembly. We also note that shape-changing colloids are another scalable system where the concept of biphasic particles may be utilized to tune interparticle interaction strengths with force[36].

As our nanoparticle model is conceptual, we shall end our discussion by presenting a possible path, scaffolded DNA origami, to expand this model into synthetic systems. In these DNA structures, stiff double-stranded DNA (dsDNA) and flexible single-stranded DNA (ssDNA) components are integrated to create mechanical devices capable of precise motions[37]. Molecular machines fabricated in the literature, i.e., hinges and sliders and especially scissors, share many structural similarities with our X-shaped nanoparticle[38]. In Supplementary Text, we provide a component-by-component analysis of the X-shape nanoparticle and compatible DNA origami structures for each component, see Supplementary Fig. 3. To summarize, stiff members, hinges, switches, and binding sites are four crucial components of our design. Stiff members can be created by dsDNA bundles and hinges can be created by connecting stiff members end to end with ssDNA connectors. The switch potential could be generated by using a hairpin loop, which would open up when the force is adequately high. To create the primary and SBSs, the tips of stiff arms can be functionalized with shorter ssDNA molecules, referred to as "sticky ends", which would facilitate interparticle binding[39]. With this modification, the sticky ends could bind one particle to another by forming duplexes with complementary sticky ends on other particles, which can create dimers and fibers that are observed in our simulations. The kinetics of the sticky ends and the hairpin can be tuned by changing the DNA sequence, e.g., lengthening the overlap between sticky ends or hairpin loop would create more hydrogen bonds and should prolong the lifetime. Last, we note that this is only one of the paths to make catch bonds; any thermalized microparticle or granular system with constituents having shape-shifting or contact modulation capability can in principle exhibit catch bonding interfaces when properly tuned.

## Methods

**Molecular dynamics (MD) simulations**. The MD simulations were performed using the LAMMPS software package[40]. The simulations were run in the NVT ensemble at 150 K using a Langevin thermostat with a damping factor of 100 time steps. The adopted time step of 2 fs was found to be sufficiently small to ensure stability. For tensile test simulations, force was applied after an equilibration of 5000 time steps. In each simulation, constant forces ranging from 140 to 280 pN are applied to the end points of the dimer at LA state (as shown in Fig. 1b) and the interaction energies between the nanoparticles were monitored. To obtain meaningful statistics for each force value, 10,000 trials were performed.

We used replica-exchange MD simulations to study the self-assembly of 100 open-conformation particles in a $200 \times 200$-nm$^2$ periodic boundary box. First, nanoparticles are considered as circles and placed randomly inside the simulation box without any overlaps. Next, they are given a random rotation. Multiple simulations (10 replicas) covering a temperature range of 150–180 K are performed in parallel. Simulations were run for $10^7$ time steps. Exchanges between adjacent replicas were attempted every $10^4$ time steps. The temperature distribution was chosen such that a constant acceptance ratio of 20% occurred between all replicas. The analysis of the simulations was carried out over frames of the baseline trajectory, i.e., replica corresponding to 150 K.

## Data availability

The authors declare that all data supporting the findings of this study are available within the paper and its supplementary information files or from the corresponding author upon reasonable request.

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

## Acknowledgements
This research was sponsored by Office of Naval Research Early Career Award (PECASE, grant # N00014-16-1-3175).

## Author contributions
K.D. and S.K. conceived the design and the theory. K.D. performed the computational experiments. All authors contributed to the discussion and the drafting of the paper.

## Competing interests
The authors declare no competing interests.
