## [Peer Review File · Nature Communications]

Reviewers' comments:

Reviewer #1 (Remarks to the Author):

Comments on "Self-strengthening biphasic nanoparticle assemblies with intrinsic catch bonds":

The authors propose a simple x-shaped particle construct for achieving catch bond like response. The presented construct is based on certain conceptual considerations that have been used in earlier works from the group for design of tweezer-like structures that show catch bond like response. The claim in this article is that the key features associated with catch bonds are (1) existence of two conformational states (2) force induced transitions between states and (3) increase in affinity by formation of new interactions between certain parts of the construct because of structural changes. The construct is composed of primary binding sites, switch and secondary binding sites with their respective features chosen such that they satisfy the specified key features associated with catch bonds. The authors support their claim that x-shape particle construct exhibits catch bond features using simulation studies on dimers formed by construct. They also present simulation studies of self-assembly of the x-shaped construct indicating weakening of the catch bond like response with fiber formation due to progressive binding between dimers. They propose use of multiple fibers in parallel for tethering two surfaces to obtain increased lifetime.

1) The authors claim the construct is conceptual. There exist conceptual models in literature for catch bonds behavior based on force induced switching between deep and a shallow energy wells described elsewhere (Pereverzev et al Phys Rev E., 79, 2009, Prezhdo and Pereverzev, Acc. of Chem Res, 42(6), 2009, Pereverzev et al Bio Phys J., 101, 2011). It is essential that the authors refer to these earlier works and provide a clear account of how the present conceptual construct is different from that of the existing conceptual models. Furthermore, the x-shaped particle construct can be considered to be of added value only if it captures specific features that have not been captured by earlier frameworks. Presently a complete discussion pertaining to the relevance of the construct in relation to that of the existing literature is missing in the manuscript.

2) The formal results obtained from conceptual models available in literature (Pereverzev et al) have been applied to well-studied experimental catch bond systems. On the contrary, the conceptual construct presented here refers to potential synthetic constructs that can be built to achieve such catch bond response and does not attempt to apply the construct to explain any observed behavior. The approach in the present work is speculative and does not merit serious consideration unless the results obtained are compared to those obtained from a similar synthetic construct.

3) Magnitude of applied force and force history are both important aspects under consideration for study of force-induced changes in lifetime of bonds (Evans et al, Proc. Natl. Acad. Sci., 101(31), 2004, Lin et al, Phys. Rev Lett., 98, 2007, Maitra and Arya, Phys. Rev. Lett, 104, 2010). In particular, one of the key features of catch bonds is that when force is increased linearly they may rupture at low or high force, but not in between, and a faster loading rate causes a sudden switch to

the high-force ruptures (Wendy Thomas, *Ann. Rev. Biomed. Eng.*, 10, 2008). The information on effect of loading rates on the response of the construct is essential for determining the catch bond utility of the construct. The present set of simulations are limited to studying effect of magnitude of applied force and do not seem to provide the necessary information on effect of force history.

4) In the present work Morse potentials are used to represent primary, switch and secondary binding site interactions. The authors have provided a rational for choice of the width of the well for the different interactions. However, a clear rational is not provided for choice of the depth of the energy well. Specifically, it would be useful to provide a rational for choosing the depth of the switch energy well to be larger than that of both the primary and secondary binding sites. It is also important to address the question whether such a choice would lead to equilibrium dissociation under absence of a force ($F=0$) to be governed by the switch rather than the primary binding site.

5) The authors propose use of multiple fibers composed of n x-shaped particles in parallel for binding of two surfaces and provide an expression for lifetime of the resulting system. Detailed analysis on life time of parallel slip and catch bonds with uniformly distributed force are available in literature (Erdmann and Schwarz, *Phys. Rev. Lett.*, 92(10), 2004, Sun et al, *J. R. Soc. Interface*, 9, 2012). The existing studies indicate that multiple-bond clusters may break in a complicated process that involves multiple stages. The authors expression of lifetime seems to be based on an assumption that average lifetime may be obtained as a simple average over an ensemble of chains under a constant force. This might not be the case as the temporal change of force on each fiber may lead to distribution of life times that are different from that of a simple ensemble average. The authors need to support the claim of average lifetime proposed through detailed simulations on the system.

Reviewer #2 (Remarks to the Author):

This manuscript proposes a type of nanoparticle that could be used to realize catch bonds in a synthetic system and that self-assembles into a fibril. The function of the particle is demonstrated through molecular dynamics simulations. This manuscript builds directly on the authors' prior work showing that a tweezer mechanism can act as a catch bond (also demonstrated through MD with some supporting theory). This work is distinct from the authors' prior work in that the mechanism is more detailed / closer to an actual design and in that self-assembly is also demonstrated – helping to address the issue of how the catch bond mechanism would function in a real system. However, the inclusion of the switch simply as an equation rather than an explicitly created mechanism is a substantial weakness. In my opinion this greatly reduces the novelty of this work compared to the author's prior work. The authors clearly try to address this weakness (and the conceptual to real design gap more broadly) in the discussion section, however this discussion falls short because the options seem to only address part of the design requirements. This work could be at a Nature Communications level if the authors can narrow this gap between their conceptual design and a real design. This could be accomplished through (1) simulating a fully atomistic or coarse-grained version of a real option, or (2) providing a more detailed version of the current discussion section with all components from nominally compatible

options, perhaps with supporting figures. While these authors are clearly leading a new approach to catch bond design with their conceptual approach, for this work to have strong impact, it needs to take that next step and be accessible to someone who is actually going to design and synthesize a real version.

The methods for both the dimer simulations and the self-assembly simulations seem sound.

For the most part the manuscript is well written with a clear thought process about the design requirements for the nanoparticle and providing insight throughout.

- There are minor grammatical errors throughout that definitely need to be fixed for readability prior to publication. For example, “We select α and x_0 of *ESBS* to be the same with primary interaction parameters” needs to be “same as” to make sense.
- The section “Biphasic . . .” is confusing since it seems to just repeat the content of the previous section with an additional figure. This section should either be removed, integrated into the previous section, or improved.
- Although the information is in the SI, the bonded interactions within each nanoparticle are not mentioned at all in the manuscript (only the binding sites and steric repulsion are mentioned). On first read this reviewer thought that the beads were self-assembling into nanoparticles.

Reviewer #3 (Remarks to the Author):

The authors use MD simulations to show a X-shaped nanoparticle exhibiting catch bond function. They measure energy landscapes for binding/unbinding in the two conformations, establish relations for bond lifetime, and simulate filaments of up to 10 units. The catch bond concept is attractive across many fields in biology and engineering and this group demonstrates a simple solution, building on their previous work, to extend the idea to 1D filaments while laying the groundwork for assembly of multiple fibers in parallel. I just have some minor comments that should be addressed for publication

- This is a great manuscript overall and the figures are clear. Establishing the three key catch bond features will be helpful for the broad audience of this journal
- The switch is a little confusing. I think the authors could add more detail to explain how this works. Is it a hinge like the other links and kept straight when activated by the secondary binding?
- In the section titles “biphasic system with two unbinding pathways”, the low-angle energy barrier in Fig 4B is explained but there is little explanation about high angles or about Fig 4C
- Reference styles are inconsistent and need attention
- I find the supplemental figures interesting and think they should be briefly mentioned in the main text
- Page 2 last paragraph: first sentence is missing a verb

Response to Reviewers' comments:

Reviewer #1 (Remarks to the Author):

Comments on “Self-strengthening biphasic nanoparticle assemblies with intrinsic catch bonds”:

The authors propose a simple x-shaped particle construct for achieving catch bond like response. The presented construct is based on certain conceptual considerations that have been used in earlier works from the group for design of tweezer-like structures that show catch bond like response. The claim in this article is that the key features associated with catch bonds are (1) existence of two conformational states (2) force induced transitions between states and (3) increase in affinity by formation of new interactions between certain parts of the construct because of structural changes. The construct is composed of primary binding sites, switch and secondary binding sites with their respective features chosen such that they satisfy the specified key features associated with catch bonds. The authors support their claim that x-shape particle construct exhibits catch bond features using simulation studies on dimers formed by construct. They also present simulation studies of self-assembly of the x-shaped construct indicating weakening of the catch bond like response with fiber formation due to progressive binding between dimers. They propose use of multiple fibers in parallel for tethering two surfaces to obtain increased lifetime.

We thank the reviewer for his/her constructive feedback, which helped us improve the manuscript considerably. We hope that the revised manuscript adequately addresses all points raised.

1) The authors claim the construct is conceptual. There exist conceptual models in literature for catch bonds behavior based on force induced switching between deep and a shallow energy wells described elsewhere (Pereverzev et al Phys Rev E., 79, 2009, Prezhdov and Pereverzev, Acc. of Chem Res, 42(6), 2009, Pereverzev et al Bio Phys J., 101, 2011). It is essential that the authors refer to these earlier works and provide a clear account of how the present conceptual construct is different from that of the existing conceptual models. Furthermore, the x-shaped particle construct can be considered to be of added value only if it captures specific features that have not been captured by earlier frameworks. Presently a complete discussion pertaining to the relevance of the construct in relation to that of the existing literature is missing in the manuscript.

We thank the reviewer for pointing us to these prior studies and for the opportunity to clarify the novelty of our approach in the context of the state-of-the-art. The works cited focus on explaining experimental observations on well-known protein catch bonds and involve mathematical models that implicitly represent the underlying system using simplified energy landscapes. The solution of the lifetimes is also based on statistical mechanics concepts rather

than actual particle dynamics, making use of idealizations such as Bell's theorem for bond breaking. Conversely, our work presents an explicit dynamical model that provides a novel structure and mechanism for catch bonds distinct from existing systems. This is a crucial intermediate step between analytical and chemistry-specific models for the inverse problem of synthetic design. Breaking kinetics comes directly from the dynamical simulations without the need for assumptions or simplifications of the energy landscape in our model. We run molecular dynamics simulations and construct lifetimes curves ourselves rather than rely on experimental data to train a theoretical model, and present a conceptual structural design that doesn't exist in literature. We also illustrate a molecular model that is capable of self-assembly, which paves the way for scalability of catch-bond based systems. Our model provides detailed trajectories of conformational transitions and kinetics of the catch bond system rather than relying of effective rate constants to control transitions from one abstract state to another. Other than these novel contributions, the following structural features are not captured by the earlier frameworks:

- 1) Explicit structures for two conformational states
- 2) A switch mechanism to smoothly transition between states with real kinetics
- 3) Complementary interaction sites that allow self-assembly of arbitrarily long fibers
- 4) Stochastic forward and reverse transitions with visual representation of trajectories

Unlike the models mentioned, we didn't define two Bell equations for different states of catch bond. Our model dynamically transitions and forms new contacts with its targeted molecule. Thus, the reaction coordinates of the energy landscape we have created actually correspond to the explicitly modeled structural features of our system.

In conclusion, the purpose and contributions of our model and models cited are vastly different, as we are not trying to predict the lifetime curve or other features of a specific protein. We are interested in creating a novel structural model that provides insights into possible molecular designs that serve as templates for synthetic particles with intrinsic catch bonds. This work builds upon our recent publication (Keten & Dansuk, *Matter*, 2019) by implementing hidden secondary interactions and a switch mechanism in self-assembling particles that can grow into fibers.

To clarify these distinctions as the reviewer suggested, we added a statement that summarizes the gist of the argument that we presented above.

The following statement is added to the first paragraph on page 3: "So far, the catch bond research has led to the development of several theoretical models^{18,19}. Phenomenological models, such as the two state model^{20,21} and the two path model^{22,23}, can accurately represent characteristics of a wide range of experimental data sets, but provide limited information on how one should create a structure and mechanism for catch bonds, since the parameters extracted from these models does not have clear structural interpretations."

The following statement is added to the last paragraph on page 4: "We believe the proposed structure and mechanism for catch bonds is a crucial intermediate step for the inverse problem of synthetic design. Unlike the phenomenological models in the literature, our system does not

rely on experimental data to train a model and makes no assumptions on energy landscapes or bond breaking kinetics. Lifetime curves are directly obtained from MD simulations. In addition, our model displays (i) explicit structures for two conformational states, (ii) a switch mechanism to smoothly transition between states with real kinetics, and (iii) complementary interaction sites that allow self-assembly of arbitrarily long fibers, which are all structural features that are needed for synthetic design, but were not captured by the earlier frameworks. Furthermore, the molecular model is capable of self-assembly, which paves the way for scalability of catch-bond based systems.”

The last part of reviewer’s comment is directly related to the second comment that he/she made below, thus in order not to repeat ourselves, this will be addressed below under second comment.

2) The formal results obtained from conceptual models available in literature (Pereverzev et al) have been applied to well-studied experimental catch bond systems. On the contrary, the conceptual construct presented here refers to potential synthetic constructs that can be built to achieve such catch bond response and does not attempt to apply the construct to explain any observed behavior. The approach in the present work is speculative and does not merit serious consideration unless the results obtained are compared to those obtained from a similar synthetic construct.

We want to reiterate to the reviewer that synthetic molecules that fully reproduce catch bond characteristics haven’t yet been fabricated. We also have not speculated that our model applies quantitatively to any existing protein ligand complex, as this is not the intention of our work. The synthesis of novel constructs based on this work is beyond the scope of the work. Our goal is to present a computational thought experiment as a precursor to synthetic designs. We believe that not basing our system on a specific protein is not a weakness but rather a strength of our model as it points to generic features of catch bonds. Allosteric pathways of catch bond proteins are complex and multi-dimensional, and idealized mathematical models work well phenomenologically, but the exact mechanisms remain debated¹⁻³. We created simple structural models that exhibit catch bonds with greatly reduced degrees of freedom, which is much more challenging than matching a particular catch bond lifetime curve analytically, and a step closer to synthesis than idealized energy landscapes. The merit of our study is that it addresses a fundamental knowledge gap through its direct demonstration of a simple molecular mechanism to achieve catch bond functionality. This divergence from actual proteins is critical for progress in the field since how we’ll create catch bonds synthetically will have as many differences as similarities with proteins, which is often true in the path of innovation with bioinspiration (e.g. if we take flight as an example).

1. Structure-based discovery of glycomimetic FimH ligands as inhibitors of bacterial adhesion during urinary tract infection, Vasilios Kalas, Scott J. Hultgren PNAS Mar 2018, 115

2. Sauer, M., Jakob, R., Eras, J. et al. Catch-bond mechanism of the bacterial adhesin FimH. *Nat Commun* 7, 10738 (2016). <https://doi.org/10.1038/ncomms10738>
3. Transmission of allostery through the lectin domain in selectin-mediated cell adhesion, Travis T. Waldron, Timothy A. Springer, *PNAS* Jan 2009

These points are mentioned in the introduction, results (especially the Structure design section) and discussion, however the part that we add on page 4 last paragraph addresses the reviewer's concerns.

3) Magnitude of applied force and force history are both important aspects under consideration for study of force-induced changes in lifetime of bonds (Evans et al, *Proc. Natl. Acad. Sci.*, 101(31), 2004, Lin et al, *Phys. Rev Lett.*, 98, 2007, Maitra and Arya, *Phys. Rev. Lett.*, 104, 2010). In particular, one of the key features of catch bonds is that when force is increased linearly they may rupture at low or high force, but not in between, and a faster loading rate causes a sudden switch to the high-force ruptures (Wendy Thomas, *Ann. Rev. Biomed. Eng.*, 10, 2008). The information on effect of loading rates on the response of the construct is essential for determining the catch bond utility of the construct. The present set of simulations are limited to studying effect of magnitude of applied force and do not seem to provide the necessary information on effect of force history.

We thank the reviewer for this excellent suggestion. We performed the force-ramp simulations as the reviewer suggested and observed both phenomena that the reviewer described.

On page 12 we added a new section.

“Force history dependence of the system. Since the dimer can switch between and unbind at different conformations depending on the instantaneous value of the force, the catch bond lifetime depends on the loading history. Indeed, effects of loading history on catch bond lifetimes are investigated in FimH and P-selectin proteins via force-ramp experiments, where a probe is used to pull apart the protein-ligand pair under constant loading rate^{1,2}. These experiments show that protein-ligand interactions break at low or high forces but are strongest at an intermediate range of force. It is also observed that as the loading rate is increased, the breaking events occur more frequently at higher forces, thus at faster loading rates low-force rupture events are less likely to occur. To investigate these effects, we performed simulations where the tensile force on the dimer is increased at a constant loading rate, and we record the magnitude of the force when the bond breaks, F_b . The histogram of F_b for different loading rates is shown in Fig. 5. The curves exhibit a bimodal distribution, with one peak at low forces and another at high forces. This shows that, for our system, majority of the pulling trials break at low and high forces, but not at in between intermediate range, which is in agreement with force-ramp experiments. Similar to the experiments, we observe that as we increase the loading rate, we observe an increase in the high

force peak in the histogram and decrease in the low force peak. Note that the overall curves shift to the left with the increasing loading rate. This is expected since bond strengths are generally increased by the loading rate³.”

1. Mechanical switching and coupling between two dissociation pathways in a P-selectin adhesion bond, Evan Evans, Andrew Leung, Volkmar Heinrich, Cheng Zhu PNAS Aug 2004, 101 (31) 11281-11286; DOI:10.1073/pnas.0401870101
2. FimH Forms Catch Bonds That Are Enhanced by Mechanical Force Due to Allosteric Regulation Olga Yakovenko, Wendy E. Thomas J. Biol. Chem. 2008 283
3. Probing the Relation Between Force—Lifetime—and Chemistry in Single Molecular Bonds, Evan Evans, Annual Review of Biophysics and Biomolecular Structure 2001 30:1, 105-128

Figure 5. Break force histograms for X-shaped nanoparticle dimer under constant loading rate. The data is expressed as a fraction of total pulls (10,000 trials) for each of the three loading rates.

4) In the present work Morse potentials are used to represent primary, switch and secondary binding site interactions. The authors have provided a rational for choice of the width of the well for the different interactions. However, a clear rational is not provided for choice of the depth of the energy well. Specifically, it would be useful to provide a rational for choosing the depth of the switch energy well to be larger than that of both the primary and secondary binding sites. It is also important to address the question whether such a choice would lead to equilibrium dissociation under absence of a force ($F=0$) to be governed by the switch rather than the primary binding site.

We have added a section explaining our choice of selecting the switch well depth greater than the binding site well depth. As for the equilibrium dissociation being governed by the switch, we don't see how this is possible with the selected parameters in the manuscript. On the other hand, if the switch energy barrier is lower, this could influence the binding kinetics even in the absence of force. However, this is at odds with our first condition for catch bond systems, which was "At small forces most trials should result in LA unbinding of the dimers, and at large forces, most trials should transition to the HA state before LA unbinding can occur." Given that the switch energy barrier is low, switch lifetime will be shorter than LA unbinding lifetime, this condition will not be met.

The following statement is added to the first paragraph on page 8: "To satisfy the first condition, we determined that the switch should have a deeper and a broader energy landscape than the binding site landscape²⁹. At zero or small forces, the switch lifetime should be longer than the LA unbinding lifetime to have the majority of the pulling trials result in dimers separating in the LA state before transitioning to the HA state. According to Kramer's theory³⁰, $\langle \tau \rangle$ is directly proportional to $\exp[D_0]$, thus by selecting the depth of the energy well of the switch ($D_{0,S}$) to be greater than depth of the energy well of the primary binding sites ($D_{0,PBS}$), we obtain a higher switch lifetime at small forces. Note that in catch bonds, conformational changes occur over relatively long time scales in the absence of force, which is also why they are difficult to observe with MD simulations^{9,31}. This suggests that there is a large barrier to these conformational changes, which is in qualitative agreement with our choice for $D_{0,S}$. On the other hand, the LA unbinding lifetime should be longer than the switch lifetime to have majority of the pulling trials result in HA state transition at large forces. This transition between the relative lifetimes will be controlled via tuning the width of the energy landscapes of the system. Note that the tilting of the energy landscape by force is sensitive to the distance to the transition state, or the width of the landscape when well depth is kept constant. Therefore, E_S must have lower α (broader well) compared to E_{PBS} , which will result in the energy barrier of the switch landscape to decrease rapidly when greater tensile force is applied. Hence, the switch lifetime declines more sharply when subjected to force compared to the LA unbinding lifetime. Based on this condition, we have determined the parameters for E_S and E_{PBS} , which are listed in Table 1."

5) The authors propose use of multiple fibers composed of n x-shaped particles in parallel for binding of two surfaces and provide an expression for lifetime of the resulting system. Detailed analysis on life time of parallel slip and catch bonds with uniformly distributed force are available in literature (Erdmann and Schwarz, Phys. Rev. Lett., 92(10), 2004, Sun et al, J. R. Soc. Interface, 9, 2012). The existing studies indicate that multiple-bond clusters may break in a complicated process that involves multiple stages. The authors expression of lifetime seems to be based on an assumption that average lifetime may be obtained as a simple average over an ensemble of chains under a constant force. This might not be the case as the temporal change of force on each fiber may lead to distribution of life times that are different from that of a simple ensemble

average. The authors need to support the claim of average lifetime proposed through detailed simulations on the system.

As the reviewer stated we haven't tested cooperation of multiple parallel fibers, which could potentially exhibit greater complexity depending on the boundary conditions used. To keep the manuscript focused on single fiber behavior, we have removed this portion from the manuscript.

Reviewer #2 (Remarks to the Author):

This manuscript proposes a type of nanoparticle that could be used to realize catch bonds in a synthetic system and that self-assembles into a fibril. The function of the particle is demonstrated through molecular dynamics simulations. This manuscript builds directly on the authors' prior work showing that a tweezer mechanism can act as a catch bond (also demonstrated through MD with some supporting theory). This work is distinct from the authors' prior work in that the mechanism is more detailed / closer to an actual design and in that self-assembly is also demonstrated – helping to address the issue of how the catch bond mechanism would function in a real system. However, the inclusion of the switch simply as an equation rather than an explicitly created mechanism is a substantial weakness. In my opinion this greatly reduces the novelty of this work compared to the author's prior work. The authors clearly try to address this weakness (and the conceptual to real design gap more broadly) in the discussion section, however this discussion falls short because the options seem to only address part of the design requirements. This work could be at a Nature Communications level if the authors can narrow this gap between their conceptual design and a real design. This could be accomplished through (1) simulating a fully atomistic or coarse-grained version of a real option, or (2) providing a more detailed version of the current discussion section with all components from nominally compatible options, perhaps with supporting figures. While these authors are clearly leading a new approach to catch bond design with their conceptual approach, for this work to have strong impact, it needs to take that next step and be accessible to someone who is actually going to design and synthesize a real version.

We thank the reviewer for his/her constructive feedback and suggestions. We feel that it has helped us improve the manuscript considerably and hope that with the new revisions the manuscript will be satisfactory.

1) Switch mechanism explained

Perhaps there was a misunderstanding in this section. Like our prior work, the switch is explicitly modeled rather than with an equation. We are using the switch mechanism that is used in our prior work. Switch is a two-member hinge and its free ends interact with a Morse potential which

keeps them at a certain distance in the absence of force. The force breaks this weak interaction, which triggers the conformational change. These transitions are explicitly modeled using molecular dynamics simulations with the potentials described in the paper. The sole analytical portion of this work is related to predicting lifetime of self-assembled fiber systems.

In order to make the explanation of the switch mechanism clearer the following statement is revised in first paragraph on page 7: “The switch consists of a two-member hinge with a pairwise interaction between its two free ends (blue SBS bead and black bead at the symmetry axes). In the open conformation, the switch members are angled due to the pairwise interactions as shown with black dashed lines in Fig. 1B. However, the switch interaction can break due to external forces. In that case, since there is no other force between the members, system elongates in the direction of the force and the switch members straighten and become linear (Fig. 1C).”

2) Forming the bridge between the real and conceptual design

We have found that it was not feasible to simulate catch bond behavior with reasonable statistical sampling with fully atomistic model or more detailed coarse-grained models of realistic equivalent systems. To provide some context, the lifetime statistics of catch bonds follow an exponential distribution, where the mean value is equal to the variance, which makes it difficult to accurately obtain the force – lifetime curve from limited sample sets. In this article, we ran 10,000 simulations for each data point of the lifetime curves, including the new runs that examined rate dependence as requested by the reviewers. While there are CG models for DNA and other molecular systems that may be used to study these systems, the representation of the complete system with a model like oxDNA would require roughly 1000x more beads per nanoparticle, which increases the computational cost by many orders of magnitude. These computations would take many years even in the best supercomputers. This is also the reason why direct simulation of explicit catch bonds between two particles has not been published thus far, making this study being the first of its kind. We also demonstrate self-assembly capability in these systems with replica exchange MD, as well as scaling of the lifetime with size, which are again computationally intensive calculations that cannot currently be carried out with higher fidelity models.

The advantage of our system is it has very few particles, bonded interactions and we are not limited to predetermined force fields that takes a lot of computational time. The simplicity of our model enables us to look at large number of trials at different force levels at manageable timescales. The system shows the dynamic transition between two states and shows spatial relation between the shape of the system and the position of the interactions.

In this regard, we are taking the second path that the reviewer has suggested. In order to provide a direct link to a real system that can currently be synthesized, we expand on our suggestion of using DNA origami to create the various components that make up our nanoparticles. We admit that in original design didn't think deeply about synthesis considerations and instead focused on

a simple design based on our mechanical intuition. Now that we are requested to make a connection to chemical synthesis, we concede that there may be simpler ways to make catch bonds. Regardless, the discussion we added should provide adequate rationale as to how each component could be synthesized to achieve the catch bond function described in this paper. Our hope here is that experimentalists that are not bound by the original computational model we came up with and can take inspirations from this work to arrive at simpler, cost-effective designs.

The entire second paragraph of the Discussion section on page 17 is revised.

“As our nanoparticle model is conceptual, we shall end our discussion by presenting a possible path, scaffolded DNA origami, to expand this model into synthetic systems. In these DNA structures, stiff double-stranded DNA (dsDNA) and flexible single-stranded DNA (ssDNA) components are integrated to create mechanical devices capable of precise motions³¹. Molecular machines fabricated in literature, i.e. hinges and sliders and especially scissors, share many structural similarities with our X-shaped nanoparticle³². In the Supplementary Text, we provide a component by component analysis of the X-shape nanoparticle and provide compatible DNA origami structures for each component, see Fig S3. To summarize, stiff members, hinges, switches and binding sites are four crucial components of our design. Stiff members can be created by dsDNA bundles and hinges can be created by connecting stiff members end to end with ssDNA connectors. The switch potential could be generated by using a hairpin loop, which would open up when the force is adequately high. To create the primary and secondary binding sites, the tips of stiff arms can be functionalized with shorter single-stranded DNA molecules, referred to as “sticky ends”, which would facilitate interparticle binding³³. With this modification, the sticky ends could bind one particle to another by forming duplexes with complementary sticky ends on other particles, which can create dimers and fibers that are observed in our simulations. The kinetics of the sticky ends and the hairpin can be tuned by changing the DNA sequence, e.g. lengthening the overlap between sticky ends or hairpin loop would create more hydrogen bonds and should prolong the lifetime. Lastly, we note that this is only one of the paths to make catch bonds; any thermalized microparticle or granular system with constituents having shape-shifting or contact modulation capability can in principle exhibit catch bonding interfaces when properly tuned.”

In addition, new supplementary text and a new supplementary figure are added to elaborate DNA origami construct of the X-shape nanoparticle.

X-shaped nanoparticle using DNA origami

Our nanoparticle design offers simple thought experiments that generate important insights for building nanoparticles with catch bonding interfaces, however, many more steps are needed to go from this design to a synthetic system. In regard to this, we offer some rationale as to how

one might go about creating various components of the nanoparticle, taking the versatile framework of DNA origami as an example. As shown in Fig. S3, we highlight four main components of the nanoparticle: stiff members, hinges, switch and the interaction sites.

Stiff members can be created from a bundle of interconnected double-stranded DNA helices organized in a honeycomb configuration (Fig. S3B). Helix-bundles reportedly reach persistence lengths that exceed 2000 nm, thus will behave as stiff rods in the relevant scales to our system¹. These members can be connected from both ends by several flexible single-stranded DNA (ssDNA) scaffold connections arranged in a line to form the hinge rotation axis (Fig. S3C). ssDNA exhibits a persistence length of ~ 2 nm, thus it is expected the system would be compliant at the hinge joints. Moreover, it has been shown in various DNA origami structures that ssDNA hinge can rotate flexibly over a range of angles².

In our X-shape nanoparticle design, the two-membered angled switch served two purposes. First, it provided an energy barrier between two conformations. Second, it limited further deformation when system transitioned to the closed state. These effects can also be created with a DNA loop structure, i.e. a hairpin. Hairpins are ssDNA or RNA sequences with complementary base pairs. In absence of force, these base pairs bind with hydrogen bonds form a loop (Fig. S3D). When the hairpin is under force, hydrogen bonds break and cause the chain to unfold and increase its extensibility. Thus, hairpin creates an energy barrier for conformational change due to its hydrogen bonds and prevents elongation after the chain elongates to its contour length.

Lastly, DNA overhangs can be used to attain different rate kinetics in primary and secondary binding sites at the tips of the X-shaped nanoparticle. Overhangs in this case are ssDNA that have sticky ends. As shown in Fig. S3E, complementary sticky ends (shown in red) form hydrogen bonds and form the bridge between two nanoparticles. On the other hand, inner bases (shown in grey) play the role of flexible linkers. For a system to show catch bond behavior, we showed that energy landscapes of primary, secondary and switch energy landscapes differ in well depth and width. Overhangs enable us to tune energy landscapes since sticky end length and base sequence effects the strength of DNA sticky end links³. Thus, increasing the complementary bases would deepen the energy well, and the flexible linkers could offer compliance to tune the curvature of the energy landscape⁴.

Figure S3. Possible DNA origami design of the X-shaped nanoparticle. A) Four main components of the nanoparticle are stiff members, hinges, switch and the interaction sites. These components can be created with B) double-stranded DNA helix bundles, C) single-strand DNA hinges, D) DNA hairpins, E) DNA overhangs with sticky ends. The members and helical appendages are color coded according to Figure 1.

1. Kauert, D. J.; Kurth, T.; Liedl, T.; Seidel, R. Direct Mechanical Measurements Reveal the Material Properties of Three- Dimensional DNA Origami. *Nano Lett.* 2011, 11, 5558– 5563
2. A. Marras, L. Zhou, H.Su, C. Castro, Programmable motion of DNA origami mechanisms, *Proceedings of the National Academy of Sciences* Jan 2015, 112 (3) 713-718
3. E. Ban and C. Picu, Strength of DNA Sticky End Links, *Biomacromolecules* 2014 15 (1), 143-149
4. A. Maitra and G. Arya, Model Accounting for the Effects of Pulling-Device Stiffness in the Analyses of Single-Molecule Force Measurements, *Phys. Rev. Lett.* **104**, 108301 – Published 12 March 2010

The methods for both the dimer simulations and the self-assembly simulations seem sound.

For the most part the manuscript is well written with a clear thought process about the design requirements for the nanoparticle and providing insight throughout.

- There are minor grammatical errors throughout that definitely need to be fixed for readability prior to publication. For example, “We select α and x_0 of *ESBS* to be the same with primary interaction parameters” needs to be “same as” to make sense.

The errors have been addressed during the revisions.

- The section “Biphasic . . .” is confusing since it seems to just repeat the content of the previous section with an additional figure. This section should either be removed, integrated into the previous section, or improved.

We believe that it is important to show the multidimensional energy landscape of the system. Thus, this section is integrated into the previous section.

- Although the information is in the SI, the bonded interactions within each nanoparticle are not mentioned at all in the manuscript (only the binding sites and steric repulsion are mentioned). On first read this reviewer thought that the beads were self-assembling into nanoparticles.

We have added an extra panel to Figure 1A, where we show the skeleton of the nanoparticle system with its hinge locations indicated as nodes. The caption and ‘Structure design’ sections are modified accordingly.

Figure 1. Schematics of X-shaped nanoparticles. (A) Each nanoparticle is formed by rigid members connected with hinge joints (black nodes). Each nanoparticle has primary binding sites (PBS, red), secondary binding sites (SBS, blue) and switch mechanisms. Hinge-like switch is

highlighted with the dashed box. Grey beads are located on the rigid members and have steric repulsion to create inaccessible volume determining the shape of the nanoparticles.

Reviewer #3 (Remarks to the Author):

The authors use MD simulations to show a X-shaped nanoparticle exhibiting catch bond function. They measure energy landscapes for binding/unbinding in the two conformations, establish relations for bond lifetime, and simulate filaments of up to 10 units. The catch bond concept is attractive across many fields in biology and engineering and this group demonstrates a simple solution, building on their previous work, to extend the idea to 1D filaments while laying the groundwork for assembly of multiple fibers in parallel. I just have some minor comments that should be addressed for publication

- This is a great manuscript overall and the figures are clear. Establishing the three key catch bond features will be helpful for the broad audience of this journal

We thank the reviewer for his/her constructive feedback and suggestions.

- The switch is a little confusing. I think the authors could add more detail to explain how this works. Is it a hinge like the other links and kept straight when activated by the secondary binding?

In order to make the explanation of the hinge mechanism clearer, the following statement is changed in first paragraph on page 7: “The switch consists of a two-member hinge with a pairwise interaction between its two free ends (blue SBS bead and black bead at the symmetry axes). In the open conformation, the switch members are angled due to the pairwise interactions as shown with black dashed lines in Fig. 1B. However, these pairwise interaction can break due to external forces. In that case, since there is no other force between the members, system elongates in the direction of the force and the switch members straighten and become linear (Fig. 1C).”

- In the section titles “biphasic system with two unbinding pathways”, the low-angle energy barrier in Fig 4B is explained but there is little explanation about high angles or about Fig 4C

The following statement is added to the first paragraph on page 12: “Deforming the system after this point requires a large energy input, thus, there is a big surge in the transition energy below 42° . Similarly, above 50° since the distance between switch elements is less than the Morse interaction equilibrium distance of E_s ; the short range repulsion term of the potential dominates, and we see an increase in energy.”

The purpose of Fig 4C is to show that the HA unbinding pathway has a higher energy barrier than LA unbinding, which is mentioned in this section.

- Reference styles are inconsistent and need attention
- I find the supplemental figures interesting and think they should be briefly mentioned in the main text
- Page 2 last paragraph: first sentence is missing a verb

We thank the reviewer for noting these issues. All three of these comments have been addressed in the revised manuscript.

REVIEWERS' COMMENTS

Reviewer #1 (Remarks to the Author):

The reviewer would like to thank the authors for clarifying many of the points raised by the reviewer and performing the simulations showing the effect of loading rate.

The updated manuscript is well written and clear. The authors have addressed the reviewers concerns with appropriate clarifications and modifications.

I have a minor comment:

The loading rates used in the simulations seem to be quite high. It would be useful if the loading rates are compared to that of loading rates in typical force spectroscopy experiments like AFM and optical tweezers. This is likely to be of particular interest for experimental groups interested in designing synthetic catch bonds.

Thank you
Sincerely
Balaji Iyer

Reviewer #2 (Remarks to the Author):

The authors have sufficiently addressed my concerns from the initial submission.

Reviewer #1 (Remarks to the Author):

The reviewer would like to thank the authors for clarifying many of the points raised by the reviewer and performing the simulations showing the effect of loading rate.

The updated manuscript is well written and clear. The authors have addressed the reviewers concerns with appropriate clarifications and modifications.

I have a minor comment:

The loading rates used in the simulations seem to be quite high. It would be useful if the loading rates are compared to that of loading rates in typical force spectroscopy experiments like AFM and optical tweezers. This is likely to be of particular interest for experimental groups interested in designing synthetic catch bonds.

We thank the reviewer for the comments. We have added a comparison paragraph **on page 14**:

“It should be noted that the simulation loading rates are five orders of magnitude higher than the experimental loading rates of single molecule force spectroscopy experiments on catch bond systems. In our system, due to computational time restrictions, we have chosen the energy barriers of the particle interactions to be lower than the energy barriers of protein-ligand complexes to efficiently sample bond breaking events. Moreover, to reduce the noise, these simulations are performed at a lower temperature than the single molecule force spectroscopy experiments. As a result, our event lifetimes are significantly shorter than catch bond lifetimes seen in experiments. Consequently, higher loading rates have to be used to reach the critical force governing low to high affinity transition.”